# Promoting Mental Health and Wellbeing in Multicultural Australia: A Collaborative Regional Approach

**DOI:** 10.3390/ijerph19052723

**Published:** 2022-02-26

**Authors:** Ilse Blignault, Hend Saab, Lisa Woodland, Klara Giourgas, Heba Baddah

**Affiliations:** 1Translational Health Research Institute, Western Sydney University, Penrith 2751, Australia; 2South Eastern Sydney Local Health District, Multicultural Health Service, Darlinghurst 2010, Australia; hend.saab@health.nsw.gov.au (H.S.); klara.giourgas@health.nsw.gov.au (K.G.); heba.baddah@health.nsw.gov.au (H.B.); 3St George Community Mental Health, Kogarah 2172, Australia; 4South Eastern Sydney Local Health District, Population and Community Health, Darlinghurst 2010, Australia; lisa.woodland@health.nsw.gov.au; 5Centre for Primary Health Care and Equity, UNSW Sydney, Sydney 2052, Australia

**Keywords:** mindfulness-based intervention, stress management, mental health promotion, stepped care model, evaluation, migrant, Arabic speakers, Bangla speakers, Muslim, cultural adaptation

## Abstract

Migrant communities are often under-served by mental health services. Lack of community engagement results in missed opportunities for mental health promotion and early intervention, delayed care, and high rates of untreated psychological distress. Bilingual clinicians and others who work with these communities lack linguistically and culturally appropriate resources. This article reports on the implementation and evaluation of a community-based group mindfulness program delivered to Arabic and Bangla-speaking communities in Sydney, Australia, including modifications made to the content and format in response to the COVID-19 pandemic. The program was positioned within a stepped-care model for primary mental health care and adopted a collaborative regional approach. In addition to improved mental health outcomes for face-to-face and online program participants, we have documented numerous referrals to specialist services and extensive diffusion of mindfulness skills, mostly to family members, within each community. Community partnerships were critical to community engagement. Training workshops to build the skills of the bilingual health and community workforce increased the program’s reach. In immigrant nations such as Australia, mainstream mental health promotion must be complemented by activities that target specific population groups. Scaled up, and with appropriate adaptation, the group mindfulness program offers a low-intensity in-language intervention for under-served communities.

## 1. Introduction

The increasing number of international migrants and refugees worldwide presents a challenge for the delivery of health services and health promotion programs in destination countries. In 2020, there were 281 million international migrants [1], with refugees accounting for approximately 12 percent of the total [2]. Since WWII, Australia has been a major receiving country [3]. At the 2016 Census, 28% of Australia’s population were born overseas [4], a level that is higher than most countries within the Organisation for Economic Co-operation and Development [5]. Another 21% of the population had one or both parents who were born overseas [4].

In 2016, multicultural Australia was home to people with more than 300 different ancestries and speaking over 300 different languages [4]. Such diversity is recognised as a national strength [6]. It is captured in the collective term ‘culturally and linguistically diverse (CALD)’ which refers to “the non-Indigenous cultural and linguistic groups represented in the Australian population who identify as having cultural or linguistic connections with their place of birth, ancestry or ethnic origin, religion, preferred language or language spoken at home” [7] (p. 3).

Mental health is a state of wellbeing whereby individuals recognise their abilities, are able to cope with normal stresses of life, work productively and fruitfully and contribute to their communities [8]. Migrants face numerous stressors that can affect their mental health and place them at heightened risk of developing mental disorders [9]. For example, the ability to find meaningful work may be compromised by difficulty gaining recognition for educational qualifications and employment credentials [10]. Other common stressors relate to changes to traditional gender roles and intergenerational conflicts within the family [10,11], and to discrimination and social exclusion on the part of the host society [10,12]. Refugees, fleeing war and conflict in their country of origin, have experienced violence and loss and the psychological impact of an often uncertain and prolonged journey [13]. Their distress is often exacerbated by social, economic, and legal circumstances in the new country [14].

Australia has long been a leader in mental health policy and service development, with a strategy of ongoing national reform [15]. The National Mental Health Policy 2008, which embedded a whole of government approach to mental health, also embedded mental health promotion and prevention into services [15]. As defined in the NMHP, “Mental health promotion aims to maximise the ability of [individuals] to realise their potential, cope with normal stresses of life, and participate meaningfully in their communities. It also seeks to increase awareness and understanding of mental health problems and mental illness, reduce stigma and discrimination, and encourage help-seeking behaviour where this is needed” [15] (p. 13). Mental health promotion works by strengthening individuals, strengthening communities, and reducing structural barriers to health [16]. Successful programs typically involve many different government agencies and community organisations and integrate the levels of action [15,16].

In the Fifth National Mental Health and Suicide Prevention Plan 2017-22, all Australian governments committed to working together to achieve integration in planning and service delivery at a regional level [17]. The Plan extended the role of the newly established Primary Health Networks (PHNs) to provide a regionally driven stepped care approach to mental health service delivery: from promotion and prevention to early intervention, treatment, and recovery [18,19]. As service commissioners and system integrators, an important aspect of the PHN role is mitigation of identified gaps and inequities for under-served groups, including people from CALD backgrounds [20].

The stepped care model for primary mental health care is summarised in Figure 1 [21]. In Step 1 the focus is promotion and prevention for the well population, mainly publicly available information, and self-help resources. In Step 2 the focus is early intervention for at-risk groups (people with early symptoms or previous illness), mainly self-help resources, including digital mental health (online and phone support). In Step 3 the focus is access to low intensity services for people with mild mental illness through a mix of self-help resources and low intensity face-to-face services, with psychological services for those who require them. Steps 4 and 5 focus on face-to-face clinical care for people with moderate and severe mental illness.

The CALD Mindfulness Program is an ongoing program of research involving the development, implementation, and evaluation of mindfulness-based interventions (MBIs) tailored for migrant and refugee communities. MBIs, such as mindfulness-based stress reduction, can significantly alleviate depression, anxiety and stress and improve physical and psychological functioning [22,23]. The program’s genesis lies in the experience of a bilingual (Arabic/English speaking) psychologist employed at a Community Mental Health Service in the South Eastern Sydney Local Health District (SESLHD) who saw a need for an in-language mindfulness resource to use with Arabic-speaking clients.

The Arabic Mindfulness compact disc (CD) is a cultural adaptation of a resource produced by Dr Russ Harris, whose self-help books and CDs are very popular in Australia. The educational CD is 60 min in duration and contains five tracks [24]. During 2012–13, it was informally evaluated through interviews with Arabic-speaking clients who used the resource in conjunction with standard therapy. Positive feedback led to a series of formal evaluations focussing on mental health outcomes, cultural acceptability, and participant experience. The first two studies demonstrated that the Arabic Mindfulness CD was culturally and spiritually relevant and effective when used as a self-management tool in the home setting [24] and within a 5-week group program [25]. The MBI resulted in improved psychological wellbeing and was compatible with their cultural and religious practices [24,25]. The group program, which was promoted to newly arrived Arabic speaking women with refugee-like backgrounds, also provided opportunities for connecting with others and peer support [25].

Subsequently, the Central and Eastern Sydney Primary Health Network (CESPHN), commissioned the SESLHD Multicultural Health Service to deliver the group mindfulness program to CALD communities in their region. Within the stepped care model, the program corresponds to Step 3 (mix of self-help resources and low-intensity face-to-face services with psychological services as needed). The objectives were to deliver the program to Arabic speakers and Bangla speakers in the CESPHN region in order to reduce psychological distress, depression, anxiety, and stress; to provide in-language resources to support the program; and to train bilingual mental health clinicians and community workers in MBIs. The third study aimed to establish whether the group mindfulness program produced expected outcomes under normal operational conditions and to test its transferability to a second language group (Bangla) and scalability [26]. The program was shown to be culturally acceptable and effective, producing clinically and statistically significant improvements in mental health, facilitating access to mental health care and boosting mental health literacy for both language groups [26].

In March 2020, following the outbreak of COVID-19, health services ceased all face-to-face groups. Program resources were initially adapted to support the Arabic-speaking community through regular text messages that encouraged mindfulness practice at a time of changing and challenging circumstances. Additionally, the clinical lead (Hend Saab, HS) recorded a short video introducing mindfulness concepts and linking to the existing audio resources, which was disseminated through social media and community networks. The group program (Arabic and Bangla) was adapted for online delivery via videoconferencing (four sessions with a focus on stress management). It was also offered as a one-off refresher session for previous participants and as an open one-off session for new participants. Since December 2019, the mindfulness audio resources in Arabic, Bangla and English have been publicly available on the NSW Multicultural Health Communication Service website [27]. Resources for Mandarin and Nepali speakers have also been produced, with Spanish resources in development.

This article reports new findings from research with the two language groups that have been the focus of the CALD Mindfulness program to date (Arabic and Bangla), including online group outcomes, referrals to other mental health supports, diffusion through social networks, and in-language resource downloads. Additionally, it presents findings from a follow-up evaluation of the workforce capacity-building component of the broader project and interviews with community partners.

## 2. Materials and Methods

### 2.1. Regional Setting and Program Partners

Central and Eastern Sydney Primary Health Network (CESPHN) is the second largest of the 31 Primary Health Networks across Australia, with a resident population of approximately 1.6 million. The region is characterised by cultural diversity, with 40% of residents born overseas, 38% speaking a language other than English at home, and 6.9% not speaking English well or at all [28]. There is a focus on people experiencing socioeconomic disadvantage, including people from CALD backgrounds, as well as people experiencing complex health issues, poor health literacy and the impact of social isolation on health and wellbeing. Mental health is a priority [29]. Both local health districts within the CESPHN region were involved, SESLHD as the program lead and SLHD as a partner. The collaborative regional approach brought together key health services and fifteen community partners, including nine community organisations and six individuals (independent bilingual community workers or clinicians).

### 2.2. Target Groups

Arabic speakers have a significant cultural presence in Australia and in Central and south eastern Sydney where this work took place [3,29]. Bangla speakers are a new and emerging community in the region [30]. Nationally, Arabic is the third most spoken language after English and Mandarin [4]. The Arabic-speaking population is comprised of numerous cultural and ethnic communities and includes both Muslims and Christians. Members vary in country of origin, circumstances of arrival and length of residence in Australia, as well as age and education [3]. They form the majority of the refugee population [31]. Language maintenance is high [32]. The great majority of Australia’s Bangla speakers come from Bangladesh and are Muslim. Most are relative newcomers under the skilled migration program [33].

### 2.3. Group Mindfulness Program (Face-to-Face and Online)

The 4-week online program, introduced in response to the COVID-19 pandemic, was a modification of the 5-week face-to-face group mindfulness program described elsewhere [25,26]. Box 1 provides an overview of the 4-week online program. Appendix A includes the (English-language) mnemonics developed to summarise the content of each session. Appendix B provides a brief comparison of the two programs.

Box 1Overview of the Online CALD Mindfulness Stress Reduction Program.
**Group session 1: Introduction and Debriefing**
**Aim:** To discuss signs of stress and vulnerabilities experienced by participants, identify helpful and unhelpful stress responses and provide a set of motivating and practical stress management skills.**Video:** Mindfulness in Challenging Times.**Mindfulness practice:** Grounding exercise with sensory awareness.**Mnemonic:** HOPEFUL + L, a set of cognitive and practical tips to stay afloat during stressful times.
**Group session 2: Stress Experiences and Responses, Mindfulness**
**Aim:** To educate participants about stress from an evolutionary perspective, introduce mindfulness concepts and explore their cultural and spiritual applicability, and provide a set of mindfulness stress management skills.**Mindfulness practice:** Mindfulness breathing exercise.**Mnemonic**: Five ‘A’s of mindfulness—awareness, acknowledgement, actions, acceptance and appreciation.
**Group session 3: Mindfulness Based Stress Reduction Strategies**
**Aim:** To promote participants’ awareness of the interconnectedness of thoughts, feelings and behaviours, differentiate themselves from their thoughts, introduce mindfulness-based stress management skills.**Mindfulness practice:** Leaves on a stream exercise, Body scan exercise.**Mnemonic:** BE PRESENT, a set of mindfulness-based stress management skills and practices to help reduce distress.
**Group Session 4: Loving Kindness and Self-Compassion, Review**
**Aim:** To define sense of self, emphasise self-care practices through loving kindness and self-compassion and review stress management skills learnt throughout the program. To reflect and recap topics and discussions from previous sessions.**Mindfulness practice:** Loving kindness and self-compassion exercise.**Mnemonic:** REFLECTION, a program summary.

Originally derived from Buddhist practices, mindfulness has become a popular evidence-based tool for managing mental health problems in Western countries. Mindfulness is often taught through a variety of meditation exercises and MBIs usually incorporate meditation practice together with various cognitive and/or behavioural techniques [24]. The group program content, which was organised around the mindfulness audio tracks, was informed by clinical experience and knowledge of the target groups [25,26]. It drew on a range of psychological concepts and methods that fall under the umbrella of the ‘third wave’ of cognitive behavioural therapy, including mindfulness-based stress reduction, mindfulness-based cognitive therapy and acceptance and commitment therapy [34], and incorporated a strong spiritual element [24].

The first of the four online sessions was primarily designed to create an emotionally safe space for group participants to share and normalise their experiences in the context of the COVID-19 pandemic. In particular, guided discussion in the online session assisted participants in recognising helpful and unhelpful strategies they had been applying in managing pandemic-related stressors (social restrictions, including lockdown). Both programs were facilitated by a bilingual mental health clinician (psychologist) with support from a bilingual community worker.

Community partners were responsible for taking registrations, keeping attendance lists and general organisation. For the online program, only women were recruited due to high demand and gender sensitivity. Referrals for further mental health care were managed in various ways. Linguistic and cultural needs and preferences and financial and personal circumstances were taken into consideration as well as clinical needs. All participants who scored 30 or above on the K10 (likely to have a severe mental disorder) or as ‘severe’ or ‘extremely severe’ on the DASS21 were contacted by the clinical lead [35,36]. If the psychologist facilitator noticed a participant was distressed they would reach out to them. In the face-to-face groups, participants were also able to approach the facilitator privately during breaks or after the session and request assistance with referral.

### 2.4. In-Language Resources

Since the Arabic Mindfulness CD [24], a suite of audio and video mindfulness resources has been developed. These can be streamed or downloaded from the NSW Health Multicultural Health Communication Service (MHCS) website at no cost (see Appendix B for details) [27].

Audio resources (CDs and USBs (Universal Serial Bus)) were provided to group participants to assist with their ongoing mindfulness practice and later uploaded to the MHCS website. Individuals are able to use them as a self-help resource by downloading the tracks and completing the mindfulness exercises in their own time. Short videos, ‘Mindfulness in Challenging Times’, were produced to support community members during the COVID-19 lockdown. They also introduce the CALD Mindfulness Program.

### 2.5. Workforce Capacity Building

Workforce capacity building was designed to increase the reach of the CALD Mindfulness Program, with free training offered to bilingual mental health professionals, community workers and others interested in facilitating future mindfulness groups. The full-day workshop was divided into four sections: introduction to mindfulness and background to the CALD Mindfulness Program; stress management and the observing self; loving kindness and self-compassion; and managing painful emotions. Training materials included a copy of the CALD Mindfulness Program Participant Handbook, handouts, and scripts for six mindfulness exercises (developed from various resources and designed to be culturally acceptable and easy to translate).

### 2.6. Evaluation

#### 2.6.1. Group Mindfulness Program (Face-to-Face and Online)

Methods and tools used to evaluate the face-to-face group program have been published previously [25,26]. A similar approach was adopted for the online group program, with data collected by questionnaire at the first and final session. Sociodemographic items included age, gender, country of birth, years of residence in Australia, main language spoken at home, religion, education, and postcode. Participants were asked about health professionals, including mental health professionals, seen in the last four weeks. The final questionnaire assessed program experience using two questions (“What effect did the program have on your overall wellbeing?” and “Overall, how would you rate your experience of the program?”), with participants recording their responses on a 5-point Likert Scale (‘poor’ to ‘excellent’). It also included a question on skills transfer: “Over the past four weeks, have you shared your mindfulness skills with anyone? If yes, who?”

While the face-to-face program utilised two translated and validated mental health outcome measures (DASS21 and K10+), the online program employed only the K10+ to reduce the evaluation burden on participants. In addition to a global measure of psychological distress (K10), the K10+ includes four questions asking about the individual’s ability to work and carry out day-to-day activities, the number of times the person has seen a doctor or other health professional due to mental health issues, and how often physical health problems have been the main cause of these issues [37]. All other tools were translated into Arabic and Bangla by accredited translators and checked by other accredited translators and community members. The co-facilitators kept a record of group attendance and referrals made during and following the program. Participant feedback was noted verbatim if in English and translated if in Arabic or Bangla. Post program, community partners debriefed with the clinical lead, providing feedback on their experiences and observations for inclusion in the program summary and future program planning.

Analyses were conducted separately for Arabic and Bangla speakers. Sociodemographic and attendance data for the online groups were summarised using descriptive statistics. Pre- and post-measures of mental health were compared using the nonparametric sign test for paired samples (two-sided) as the data were not normally distributed, using SPSS v27 [38]. The null hypothesis was that the median difference in the pre- and post-measures would be zero.

Referrals were counted for all face-to-face and online group programs held between March 2017 and September 2021. We distinguished between referrals made for the participant themselves (if the person was referred to two separate services, this was counted as two referrals) and referrals made for a family member (e.g., a participant’s children, grandchildren, or partner). Destination of referral was classified as ‘private psychologist’, ‘General Practitioner (GP) for referral’, ‘family therapist’ or ‘other’ (e.g., private psychiatrist, domestic violence service, pain clinic, or the NSW Service for the Treatment and Rehabilitation of Torture and Trauma Survivors). Destination percentages were calculated after excluding those already receiving mental health care. Referrals that involved reconnection with a previous service provider were noted. Specific needs for referral (beyond anxiety or depression symptoms) were noted also.

We counted the number of participants who reported sharing the mindfulness skills and examined both the pattern and extent of spread. We classified reported recipients/beneficiaries according to the following categories: ‘any family’, ‘family unspecified’, ‘immediate family’ (including ‘spouse’, ‘child’, ‘parent’, ‘sibling’), ‘extended family’, ‘friend’ and ’other’. We also coded sharing with a ‘person overseas’ when it was reported. We then counted the number of participants who shared the skills across the various categories. We used the chi-square statistic to examine if language group (Arabic vs. Bangla), delivery format (face-to-face vs. online), gender (female vs. male) or age group (16–35 years vs. 36 years and over) were associated with skills transfer (not shared vs. shared); adopting a *p* value of 0.5. Further, we estimated extent of spread from the number of recipients/beneficiaries reported with each relationship category. To manage this, if a participant indicated a single person from a particular category (e.g., ‘daughter’ or ‘friend’) we counted one. If a participant indicated more than one person from a particular category (e.g., ‘sons and daughter’ or ‘friends’) we counted two, although in practice it could have been more. The resulting totals were, therefore, conservative.

#### 2.6.2. In-Language Resources

Unfortunately, we did not keep a complete record of the number of Arabic Mindfulness CDs distributed to program participants and, more widely, on request and through community events (several hundred). USBs were produced in Arabic (60), Bangla (60) and English (20). We counted the number of downloads of the online audio resources from 1 July to 31 December 2021 and video views on the SESLHD YouTube website since June 2020.

#### 2.6.3. Workforce Capacity Building

The training was evaluated through feedback forms distributed at the end of the workshops and an online follow-up survey. The feedback forms focussed on training content and delivery (process). The survey asked about key learnings and application of the learnings (outcomes and impact). Both were anonymous.

The feedback form contained 13 statements to which trainees were invited to show their level of agreement using a 5-point Likert Scale (‘strongly disagree’ to ‘strongly agree’), followed by two open-ended questions asking what they like most and what could be improved. In October 2019, people who had attended any of the first six workshops (delivered between August 2017 and May 2018) were invited by email to complete a questionnaire created on Survey Monkey. The survey contained nine questions with a mix of pre-coded and open-ended responses. Questions 1–5 covered gender, language spoken, role, primary work setting and date of training. Questions 6–7 asked about key learnings and applying the learnings. Question 8 asked how much impact the training had on their practice using a 5-point Likert Scale (‘no impact’ to ‘high impact’). Likert Scale responses and answers to the pre-coded questions were tabulated (frequencies and percentages) in Microsoft Excel. Responses to the open-ended questions were systematically coded and tabulated by Ilse Blignault (IB).

#### 2.6.4. Community Partnerships

We sought a better understanding of the role of community partners in supporting program rollout and reach through a series of interviews with facilitators, co-facilitators, and community workers. Individuals who had been involved in one or more face-to-face or online programs in the last two years were invited to take part in a 30-min semi-structured telephone interview conducted by a project officer recruited for this purpose. The interview guide contained 12 open-ended questions, two of which asked: “As an employee of [partner organisation], how do you think your contribution has supported the delivery and influenced the program outcomes?” “What were some of the challenges encountered as a partner organisation?” The interviews were not recorded; however, the project officer took extensive notes that informants were given the opportunity to review. Responses to the two questions were closely examined by HS to identify common experiences and issues, and the emergent findings reviewed at a research team meeting (IB, HS, HB (Heba Baddah) and project officer).

## 3. Results

### 3.1. Group Mindfulness Program

Between March 2017 and September 2021, 43 in-language group programs were facilitated across the region: 37 face-to-face and 5 online. They attracted a total of 489 participants, 397 of whom completed the program. Additionally, ten one-off online stress management sessions were conducted in Arabic and Bangla (results not reported here).

#### 3.1.1. Online Groups

Forty-four Arabic and Bangla-speaking women aged 16 years and over enrolled in the online program and 35 (73% of Arabic speakers and 94% of Bangla speakers) completed it. The five groups ranged in size from 7 to 11 people.

Of the 26 Arabic-speaking women recruited, most were aged 26 to 55 years. All but two were born overseas, mostly in Lebanon, Syria, or Iraq, and all but three were Muslim. Thirteen had lived in Australia for under nine years. Seventeen spoke mainly Arabic at home, the rest Arabic and English. Eighteen possessed a post-school qualification, either trade or university. Pre-intervention, 20 scored as ‘moderate’ (25–29) or ‘severe’ (30–50) on the K10. All 19 women who completed the program attended at least three of the four sessions. Reasons for dropping out included health and family issues. Lack of access to technology and unreliable internet also affected participation.

Of the 18 Bangla-speaking women recruited, we have pre-program data for only 17. Most fell into the 26–35 age group. All but one were born in Bangladesh and all were Muslim; eleven had lived in Australia for under nine years. Twelve spoke mainly Bangla at home, the rest Bangla and English. All possessed a university qualification. Pre-intervention, 11 scored as ‘moderate’ or ‘severe’ on the K10. All 17 women who completed the program attended at least three sessions. One woman dropped out for work reasons.

For both language groups, post-program measures on the K10+ showed improvement. Post-program, fewer participants scored as ‘moderate’ and none as ‘severe’ (Table 1) and there was a statistically significant reduction in psychological distress (*p* < *0*.001) (Table 2). For Arabic speakers, there was also a significant reduction in days of cutting down work due to mental health issues in the past four weeks (*p* < *0*.01) (Table 2).

All program completers reported that their experience of the program was positive. Across both language groups, 85% indicated that the effect on their overall wellbeing was ‘very good’ or ‘excellent’ and 97% rated their experience of the program as ‘very good’ or ‘excellent’. Box 2 gives examples of the feedback received from participants at the end of the program.

Box 2Participant feedback (translated from Arabic and Bangla as necessary).
**Online group 3, Arabic**
“I am aware of my thoughts and able to let go.”“My son is doing his [final exams] and I learnt to manage my anxiety so I don’t impact him.”

**Online group 4, Arabic**
“I am very happy to have found this program in my language and culture. I am able to better understand and relate to the topics.”

**Online group 5, Bangla**
“I used to live in the past I now live in the moment and I enjoy it.”“I have learnt a lot and I am applying them in my daily life.”“This is very much needed in our community.”

**Online group 6, Arabic**
“I saw a psychologist but never benefitted like this, despite this is being virtual.”


#### 3.1.2. Referrals

Across the face-to-face and online programs combined, 302 Arabic and 187 Bangla speakers were recruited, including 444 women and 45 men. The Bangla participants tended to be younger than the Arabic speakers, most of whom were aged 36–55 years with a sizeable number aged 56–55 years. They had also spent less time in Australia on average (Bangla 8.6 years vs. Arabic 20.3 years).

At program commencement, 8.7% of face-to-face and 15.9% of online program participants were already receiving mental health care. In both delivery formats, Arabic speakers were more likely than Bangla speakers to be receiving such care (Table 3). The overall comparison by language was statistically significant: 9.6% Arabic vs. 4.3% Bangla, X^2^ (1, N = 489 = 4.68, *p* < 0.05).

As a result of all group programs, an additional 106 referrals were made for specialist care (Table 3), including 11 referrals for a participant’s family member. In the two Bangla online programs, half the participants were referred. Overall, referral was significantly more likely from the online groups than face-to-face groups—43.2% online vs. 21.7% face-to-face, X^2^ (1, *N* = 452) = 8.79, *p* < 0.01). Over three-quarters (78.3%) of all referrals were to a private psychologist due to language needs. Four participants were reconnected with a service provider whom they had previously seen, usually a private psychologist. Six were referred to a family therapist (psychologist or mental health social worker) and four were referred to their GP to obtain a psychiatrist/psychologist referral. Sixteen were referred elsewhere: to a general health, social service, or community organisation (Table 3). Specific needs for referral included relationship issues, trauma, grief, and child specialist.

#### 3.1.3. Skills Transfer

Overall, 95.2% of participants who completed the face-to-face or online program reported sharing the mindfulness skills they had learned with others in their social circle. We estimated that, collectively, the 397 participants shared the mindfulness skills with at least 922 other people, an average of 2.3 people each. Relatives were the most common recipients or beneficiaries. Over three-quarters (78.1%) of program completers reported sharing with a family member (Table 4). A minority (18.6%) simply indicated family without specifying the nature of the relationship. When this was stated, 75.8% indicated immediate family (usually spouse or child) and 6.8% indicated extended family. Just under half (45.8%) of participants reported sharing with a friend and 8.1% with another person (e.g., neighbour or colleague). Fourteen people shared with someone overseas.

The percentage of participants who reported sharing their new skills did not differ significantly by language (93.8% Arabic vs. 97.4% Bangla), group format (95.0% face-to-face vs. 97.1% online), gender (95.4% female vs. 93.1% male) or age group (96% 16–35 years vs. 94.7% 36+ years). However, Bangla speakers were significantly more likely than Arabic speakers to share with their spouse, 47.0% vs. 22.9%, X^2^ (1, *N* = 378) = 24.02, *p* < 0.001); while Arabic speakers were more likely to share with their child, 37.0% vs. 15.9%, X^2^ (1, *N* = 378) = 19.80, *p* < 0.001). Similarly, younger participants (16–35 years) were significantly more likely than older participants (36+ years) to share with their spouse, 44.4% vs. 25.2%, (1, *N* = 378) = 13.04, *p* < 0.001); while older participants were more likely to share with their child, 39.7% vs. 10.4%, X^2^ (1, *N* = 378) = 37.57, *p* < 0.001).

### 3.2. In-Language Resources

The web-based in-language audio and video mindfulness resources expanded the CALD Mindfulness Program’s reach within Australia and internationally. Table 5 shows how many times the different audio tracks were downloaded from 1 July to 31 December 2021.

Since June 2020, the ‘Mindfulness in Challenging Times’ video in Arabic has received 2837 views and the English video 1275 views. Since October 2020, the Bangla video has received 1006 views.

### 3.3. Workforce Capacity Building

Between August 2017 and December 2019, seven full-day workshops were offered across the region, attracting a total of 83 participants, including a few from outside the region connected through professional networks. Another workshop was run in June 2021 with 15 participants (results not included below).

The great majority (88.0%) of trainees were women; 45.8% were bilingual community workers and 35.0% were bilingual mental health professionals, mostly psychologists and social workers. Arabic (49.4%) and Bangla (24.1%) were the most commonly spoken community languages. Other trainees spoke Cantonese, Greek, Hindi, Indonesian, Macedonian, Mandarin, Nepali, Russian, Sinhala, Spanish, Tamil, or Urdu. Eighty-two (99%) of the trainees provided post-workshop feedback, all of whom found the training engaging and the content both practical and relevant.

Seventy-one of the trainees from the first six workshops were emailed the follow-up survey, the others being uncontactable. Forty-five (63%) responded. Comparison on gender, role and language spoken suggested that the follow-up sample was reasonably representative of everyone trained. Key learnings related to core mindfulness concepts and techniques and how cultural and religious tailoring of MBIs enhances acceptability to CALD communities. Nearly three-quarters (73%) of respondents indicated that they had applied mindfulness skills for self-care. Both bilingual mental health professionals and bilingual community workers had facilitated mindfulness groups, with 19 (42%) facilitating at least two groups. They incorporated mindfulness into their clinical/counselling sessions and community wellbeing programs, and as an adjunct to mental health care and other interventions. All indicated at least ‘moderate impact’ on their practice and 60% ‘significant impact’ or ‘high impact’.

### 3.4. Community Partnerships

We identified 17 people who met the eligibility criteria from the 15 community partners and were able to interview all but one: 10 from the Arabic speaking community, 4 from the Bangla speaking community, and 2 English-speakers who were employed by a community partner (Sydney Multicultural Community Services and the NSW Refugee Health Service). Twelve were community organisation staff while the other four, including two psychologists, worked independently. Thirteen of the informants had attended the 1-day training on MBIs for CALD communities.

All community partners reported that they were kept busy before, during and after the program, with completion of tasks often taking longer than expected. Being aligned with the organisation’s operational plan and having committed and supportive management and dedicated staff (bilingual if available, paid or volunteer) and resources (venue, equipment, and refreshments) were important enablers of success. During the program, depending on their role, the community partners facilitated, co-facilitated, or provided support to the group by encouraging participants to ask questions, contributing to the group discussion, and helping to explain the presented concepts using culturally and religiously appropriate anecdotes. The presence of a familiar face also made participants feel comfortable. The bilingual community workers provided assistance with completing the evaluation measures in the first and final sessions when required, and technical support for the online groups.

Common recruitment challenges experienced by the partners included promoting and explaining mindfulness, which was a novel concept for most people, mental health stigma and geographic restrictions under the funding agreement. Recruiting homogeneous groups (similar age, gender, and level of education) presented an additional challenge. It took time to educate and reassure community members who were hesitant to discuss mental health-related issues outside their family by emphasising group confidentiality. Once the program was established, word-of-mouth proved to be the most effective means of promotion and limiting the number of participants in the group was sometimes a problem. Adhering to the program’s geographical boundaries resulted in difficult conversations for community partners and led to feelings of abandonment and marginalisation among community members who lived outside the CESPHN region. All activities took time. Maintaining weekly contact between sessions to keep participants motivated and follow-up calls to gauge their experience with the program and its impact on their day-to-day lives could involve several phone calls. Helping participants to complete relevant paperwork and supporting online participants with technical issues added an extra burden on under-resourced and already stretched individuals and community organisations.

It was apparent that relationships were fundamental to successful community engagement and program outcomes. ‘Trust’ was mentioned repeatedly, e.g., “Having an established relationship with the community and having their trust [in] providing beneficial programs” and “Participants trusted the facilitator and felt more comfortable to seek support following the program”. Community partners commented on the professionalism and expertise of the clinician facilitators from the same cultural and linguistic background and their skill in tailoring the program to make the content relevant and accessible, with consideration of each group’s unique needs (often conveyed by partner organisation to facilitator).

## 4. Discussion

The population health framework underpinning Australian mental health policy and practice recognises that there are a complex range of determinants and consequences of mental health and illness across diverse population groups [15]. Mainstream or general population initiatives must be consolidated, expanded, and complemented by activities that target specific groups [15]. As in other immigrant nations [10,39], Australia’s CALD communities are typically under-served by both primary and specialist mental health services [9,40,41]. The CALD Mindfulness Program seeks to address some of the barriers to mental health care (including self-care) through a suite of interventions designed to support de-stigmatisation, assist people to cope with negative experiences and stressful situations, and facilitate their access to professional mental health care when needed. Over the past 5.5 years, 302 Arabic-speaking and 187 Bangla speaking adults, mostly women, have participated in the 5-week face-to-face and 4-week online group programs. Both language groups, in both delivery formats, showed clinical and statistical improvements in mental health outcomes. Across all programs, 21.6% of those recruited were referred for further care.

Achieving high-levels of recruitment, retention, and adherence to protocol in community-based interventions is challenging for mainstream health services, particularly when dealing with marginalised populations [42]. Trauma-informed practices and an attitude of cultural humility can facilitate access to mental health care for minorities in a multicultural society [43]. Having a culturally competent research team and program staff who are embedded in the community and belong to the target community, who possess good interpersonal skills and are well trained is critical [42,44]. In the group programs, integration of spiritual or religious illustrations and anecdotes was pivotal to explaining mindfulness concepts to participants who were mostly of Islamic faith, facilitating their understanding and engagement. If presented appropriately, mindfulness-based approaches are very feasible and highly resonant with Islamic thought and practice [45].

Australia’s multicultural health and community workforce is an important national resource. By training 98 bilingual mental health professionals and community workers in MBIs and cultural tailoring, the CALD Mindfulness Program has strengthened individual, organisational and community capacity to respond to mental health issues in CALD communities in the CESPHN region. In the follow-up survey, 63% of respondents indicated that the training had positively influenced their practice. The fact that 73% of respondents reported applying mindfulness skills for self-care provides further evidence of relevance and cultural acceptability. Over the course of the CALD Mindfulness Program, the bilingual (Arabic/English) clinical lead has supervised three bilingual (Arabic/English) psychology interns.

Community partners have played a major role in promoting the program, engaging individuals from the target communities, and encouraging weekly attendance and mindfulness practice. Interviews with 16 community partners reinforced the importance of weekly contact by a trusted person for retention. Shared culture and language convey cultural safety which has been defined by as “an environment that is spiritually, socially and emotionally safe for people, where there is no assault, challenge, or denial of their identity, of who they are and what they need” [46] (p. 213). Trust between community partners, community members and program providers formed the basis for the program and was reinforced through program processes and outcomes.

Culturally competent mental health promotion and care must address the social and cultural determinants of health [16,47]. Since 2020, the usual stressors associated with migration and settlement have been compounded by the COVID-19 pandemic. The impact of lockdown-related unemployment, school and business closure and social disconnection has disproportionately affected already vulnerable and marginalised populations, including CALD communities and people on low incomes [48]. An early Australian study found that overseas-born respondents were more likely to report clinically significant levels of anxiety [49]. Concurrently, the pandemic has accelerated the use of digital health interventions and telemedicine as a means of supporting mental health and wellbeing [50,51], including technology assisted MBIs [52,53].

Compared to face-to-face, more of the online participants possessed post-school qualifications (62.3% vs. 81.4%, respectively). It is likely that education and digital literacy account for some of the difference in retention between Arabic and Bangla speakers in the online program. While most are keen to return to the face-to-face format, it is clear that the online format, or possibly a blended format, has a place in responding to mental health needs, particularly for Bangla and other language groups who embrace this technology. Through sharing experiences of the pandemic, wellbeing was nurtured, and cultural identities and connections celebrated. A place (actual or virtual) that is respectful, engaging, and supportive is more likely than a mainstream service to be used by minority groups and those most in need [54].

Although online interventions typically have relatively high attrition [55], the online mindfulness group program had an overall retention rate of approximately 80%, comparable to the face-to-face program [26]. We attribute this exceptional result to the ‘human element’, i.e., the encouragement provided by the facilitators and community partners, as well as the supportive and culturally safe group format [54]. Support for this conclusion is provided by a randomised controlled trial of 4-week internet-based MBI with Chinese university students. Participants who received peer counsellor support demonstrated significantly less attrition and more course completion and reported significantly greater pre-post improvements in daily stress ratings and depression than those in the no support condition [55].

A minority of participants (8.7% face-to-face and 15.9% online) were already receiving mental health care. In both delivery formats, Arabic speakers were more likely than Bangla speakers to be receiving such care. The Arabic community in Sydney (and Australia generally) is much larger and longer established, so has greater knowledge of the health system and more resources and networks to draw upon, including a cohort of bicultural/bilingual health professionals who have been educated and trained in Australia. As a result of all group programs, an additional 106 referrals were made for specialist mental health care, the majority to bilingual psychologists in private practice: an indicator of previously unmet need.

In addition to program participants who benefited directly through a process of dissemination (i.e., active and planned efforts to encourage target groups to adopt an innovation), many other people benefited through a process of diffusion (i.e., passive spread through social networks) [56]. As participants experienced the benefits of mindfulness and other mental health self-help strategies for themselves, they were eager to share the skills and knowledge with others. Sharing of positive experiences resulted in increased demand for the program and likely reduced mental health stigma, although we did not measure this. The online resources, available in Arabic, Bangla and English and an increasing number of other community languages, are regularly downloaded.

Our research has demonstrated that the group mindfulness program provides an effective and culturally acceptable low-intensity mental health intervention for the target communities as is capable of integration into the wider health system [26]. Geographical restrictions associated with the current funding model present a predicament for implementation and further scaling up. Service boundaries defined by government departments do not necessarily align with community expectations. For many CALD communities, sense of identity and community is tied to shared experiences, practices, language, beliefs, and history, rather than geographical boundaries, and members will often travel to access programs and services aligned with their cultural values [3,9]. The mismatch between these understandings of community (a geographical place versus a group of people who depend on and interact with each other and with their environment [57]) is a barrier to identifying and responding to community needs in an effective, efficient, and meaningful way. The challenge ahead lies in extending this collaborative regional approach, which brings together health services and community partners, to other regions within the metropolitan Sydney and more broadly. This is currently hampered by a lack of mechanisms to scale up effective interventions for under-served populations at state and national levels.

This research has strengths and limitations. On the strengths side, the large multicomponent intervention, which was refined and extended over several years, had evaluation embedded across all aspects and employed mental health outcome measures translated and validated with Arabic and Bangla speakers. Referral numbers, based data collected over 43 group programs that attracted 489 participants, are likely to reflect population need. Retention and program adherence were high in both the face-to-face and online formats. We achieved a good response to the follow-up survey of trainees from the mindfulness workshops and the community partner interviews. Limitations include the small number of participants (all female) in the pre-post study of the online program. In the absence of a control group, it is impossible to conclusively demonstrate causality; however, the results are consistent with a pre-post study with a wait-list control group for the face-to-face program [24]. We did not assess understanding or application of mindfulness using a validated questionnaire and did not conduct a follow-up. Importantly, CALD communities are not homogeneous, and our results are from Arabic and Bangla speakers in the CESPHN region, mostly women. Further research is required to extend the group program to men and young people and to determine its suitability, with appropriate adaptation, for use with other language groups and in other settings.

## 5. Conclusions

The CALD Mindfulness Program, which was designed to address the mental health needs of under-served CALD communities in metropolitan Sydney, has proved very successful. This innovative community-based program, with its emphasis on promotion of mental health and wellbeing, sits well within Australia’s stepped care model for primary mental health care. The in-language and culturally tailored mental health self-help resources, and low-intensity interventions delivered face-to-face and online through a collaborative regional approach, were readily adopted by participants from the target communities (Arabic and Bangla speakers) and shared with others in their family and community networks. The program provides a non-stigmatising soft-entry point to mental health services for the Arabic and Bangla communities who live in the CESPHN region. Future directions include extension of the CALD Mindfulness Program to other language groups, further exploration of technology mediated MBIs, and further integration of the program across the health system.

## Figures and Tables

**Figure 1 ijerph-19-02723-f001:**
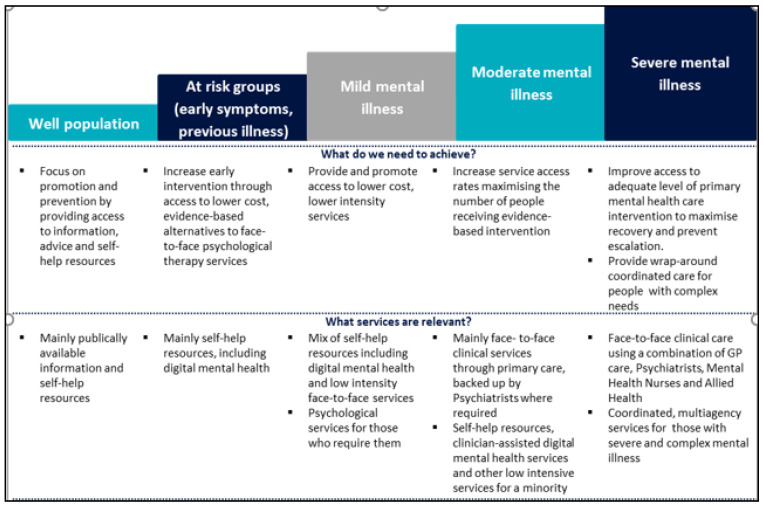
Stepped care model for primary mental health care service delivery [21].

**Table 1 ijerph-19-02723-t001:** Pre-post comparisons on the K10 category for program completers.

K10 Category	Arabic Speakers (*N* = 18)		Bangla Speakers (*N* = 17)	
	Pre-Program	Post-Program	Pre-Program	Post-Program
Well (10–19)	0	4	1	11
Mild (20–24)	4	10	5	5
Moderate (25–29)	6	4	8	1
Severe (30–50)	8	0	3	0

**Table 2 ijerph-19-02723-t002:** Pre-post comparisons on K10+ scores for program completers.

Language Group Variable	Mean Score		Sign Test for Pre-Post Change	
	Pre-Program M (SD)	Post-Program M (SD)	z	1-Sided *p*
Arabic speakers (*N* = 18)				
K10 score	28.6 (4.8)	21.3 (3.1)	−4.24	<0.001
Q11. Days of inability to work due to mental health issues in past four weeks	0.9 (1.5)	0.7(1.4)	−1.34	0.180
Q12. Days of cutting down work due to mental health issues in past four weeks (apart from days in Q11)	5.8 (6.2)	4.7 (5.5)	−2.89	<0.01
Bangla speakers (*N* = 17)				
K10 score	26.1 (4.6)	18.5 (3.5)	−4.12	<0.001
Q11. Days of inability to work due to mental health issues in past four weeks	0.4 (1.2)	0.2 (0.5)	−0.58	0.564
Q12. Days of cutting down work due to mental health issues in past four weeks (apart from days in Q11)	4.9 (8.1)	3.3 (3.9)	0	1

**Table 3 ijerph-19-02723-t003:** Participants already receiving mental health care and referrals made.

Group Language and Format	Participants Recruited	Receiving Mental Health Care ^a^	Referrals				
			Private Psychologist ^b^	Family Therapist ^b^	GP for Referral ^b^	Other Service ^b^	Total ^b^
	*N*	*n* (%)	*n* (%)	*n* (%)	*n* (%)	*n* (%)	*n* (%)
Arabic							
Face-to-face	276	22 (8.0)	45 (17.8)	4 (1.6)	3 (1.2)	8 (3.1)	60 (23.6)
Online	26	7 (26.9)	4 (21.1)	0	0	3 (15.7)	7 (36.8)
Total	302	29 (9.6)	49 (17.9)	4 (1.5)	3 (1.1)	11 (4.0)	67 (24.5)
Bangla							
Face-to-face	169	8 (4.7)	22 (13.7)	2 (1.2)	1 (0.6)	5 (3.1)	30 (18.6)
Online	18	0	9 (50.0)	0	0	0	9 (50.0)
Total	187	8 (4.3)	31 (17.3)	2 (1.1)	1 (0.6)	5 (2.8)	39 (21.8)
Overall							
Face-to-face	445	30 (8.7)	67 (16.1)	6 (1.4)	4 (1.0)	13 (3.1)	90 (21.7)
Online	44	7 (15.9)	13 (35.1)	0	0	3 (8.1)	16 (43.2)
Total	489	37 (7.6)	83 (18.4)	6 (1.3)	4 (0.9)	16 (3.5)	106 (23.5)

^a^ Percentage of all participants recruited. ^b^ Percentage of participants recruited who were not already receiving mental health care.

**Table 4 ijerph-19-02723-t004:** Sharing of skills by recipient for program completers.

Group Language and Format	Program Completers	Recipient		
		Family	Friend	Other
	*N*	*n* (%)	*n* (%)	*n* (%)
Arabic				
Face-to-face	224	170 (75.9)	91 (40.6)	18 (7.4)
Online	18	13 (72.2)	9 (50.0)	0
Total	242	183 (75.6)	100 (41.3)	18 (7.4)
Bangla				
Face-to-face	138	117 (84.8)	72 (52.2)	13 (9.4)
Online	17	10 (58.8)	10 (58.8)	1 (5.9)
Total	155	127 (81.9)	82 (52.9)	14 (93.3)
Overall				
Face-to-face	362	287 (79.3)	163 (45.0)	31 (8.6)
Online	35	23 (65.7)	19 (54.3)	1 (2.9)
Total	397	310 (78.1)	182 (45.8)	32 (8.1)

**Table 5 ijerph-19-02723-t005:** Mindfulness audio resources accessed from 1 July to 31 December 2021.

Track Title	Downloads		
	Arabic	Bangla	English
Introduction	36	10	20
Grounding Exercise with Sensory Awareness	54	5	52
Mindful Breathing	64	6	61
Leaves on a Stream	40	8	28
Body Scan	53	8	56
Practicing Loving Kindness and Self-Compassion	32	13	48
Sitting with Difficult Emotions	20	20	113

## Data Availability

The data sets are not publicly available as they contain information that could potentially re-identify individuals but are available from LW upon reasonable request and with relevant ethical approval. Program materials are available from HS.

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
