# Peer review of "Promoting Mental Health and Wellbeing in Multicultural Australia: A Collaborative Regional Approach"

_ijerph, 2022, doi:10.3390/ijerph19052723_

Round 1

Reviewer 1 Report

Some variables are ordinal scale, so, aritmethic operations should not be  performed with these data (such as mean or s.d. calculus.

Again, if there are several categorical variables used as control variables (pre/post treatment, langue, K10,...) while the response variables are ordinal, so a non-parametric analysis of variance could have been carried out.

Some references about the software used should be welcome. 

Author Response

REVIEWER 1

Comment

Some variables are ordinal scale, so, arithmetic operations should not be performed with these data (such as mean or s.d. calculus).

Response

In Table 1 on p. 9, we report pre-post comparisons on K10 category (well, mild, moderate and severe). This is an ordinal variable and we have not reported means or standard deviations. In Table 2 on p. 9, we report pre-post comparisons on K10 score, days of inability to work and days of cutting down work. These are metric variables, thus we have calculated and reported means and standard deviations.

Comment

Again, if there are several categorical variables used as control variables (pre/post treatment, langue, K10...) while the response variables are ordinal, so a non-parametric analysis of variance could have been carried out.

Response

As in our previous publication reporting on the roll-out of the face-to-face group mindfulness program to Arabic and Bangla speakers across CESPHN (Blignault et al., 2021), we analysed and presented mental health outcomes data for the online groups separately by language group.  We see this as of value to readers (researchers and practitioners) given the dearth of mental health research and outcome data on these groups.

The remainder of the statistical analyses, including referrals and skills transfer, were based on combining data from all groups conducted i.e. both language groups and both delivery formats. We used the chi-square statistic to examine if language group (Arabic vs Bangla), delivery format (face-to-face vs online), gender (female vs male) or age group (16–35 years vs 36 years and over) were associated with skills transfer (not shared vs shared); adopting a p value of 0.5.  We have now clarified this in the Methods – See lines 298–301.

In addition, finding that Arabic speakers were twice as likely as Bangla speakers to be already receiving mental health care, we used the chi-square statistic to examine if this association was statistically significant; adopting a p value of 0.5. Once again, the response variable is nominal (not receiving care vs receiving care), thus employing the non-parametric chi-square test of independence is appropriate (McHugh, 2013).

McHugh, M.L. The chi-square test of independence. Biochemia medica 2013, 23(2), 143–149. https://doi.org/10.11613/bm.2013.018

Comment

Some references about the software used should be welcome.

Response

We have added a reference for SPSS – See line 279 and reference list, but not for Microsoft Excel as this is now ubiquitous computer software.

Additional reference: IBM SPSS Statistics for Windows, Version 27. IBM Corp: Armonk, NY, USA.

Reviewer 2 Report

It was a pleasure reading the authors' paper in a such rapidly developed area. The paper is very well written and provides detailed information on the implementation and evaluation activities of a community-based group mindfulness program adapted in response to the Covid-19 pandemic. Based on the content of the programme and the authors' explanation about its origins being  Russ Harris self books it seems that at least part of the program draws from acceptance and commitment therapy or more broadly from third wave CBT approaches and mindfulness-based stress reduction. Is that correct?

It would be beneficial I think to add just a paragraph or less maybe about those theoretical underpinnings of the program theory, how they were combined and how they went around selecting certain mindfulness exercises. As it is highly likely these details have been published already elsewhere it will be adequate if authors just added a reference clearly stating that it provides those details and maybe include in the manuscript a sentence summarising a couple of points.

Author Response

REVIEWER 2
Comments

It was a pleasure reading the authors' paper in a such rapidly developed area. The paper is very well written and provides detailed information on the implementation and evaluation activities of a community-based group mindfulness program adapted in response to the Covid-19 pandemic. Based on the content of the programme and the authors' explanation about its origins being  Russ Harris self books it seems that at least part of the program draws from acceptance and commitment therapy or more broadly from third wave CBT approaches and mindfulness-based stress reduction. Is that correct?

It would be beneficial I think to add just a paragraph or less maybe about those theoretical underpinnings of the program theory, how they were combined and how they went around selecting certain mindfulness exercises. As it is highly likely these details have been published already elsewhere it will be adequate if authors just added a reference clearly stating that it provides those details and maybe include in the manuscript a sentence summarising a couple of points.

Response

We have added additional details as suggested – See lines 178–182.

Originally derived from Buddhist practices, mindfulness has become a popular evidence-based tool for managing mental health problems in Western countries. Mindfulness is often taught through a variety of meditation exercises and MBIs usually incorporate meditation practice together with various cognitive and/or behavioural techniques [24]. The group program content, which was organised around the mindfulness audio tracks, was informed by clinical experience and knowledge of the target groups [25,26]. It drew on a range of psychological concepts and methods that fall under the umbrella of the ‘third wave’ of cognitive behavioural therapy, including mindfulness-based stress reduction, mindfulness-based cognitive therapy and acceptance and commitment therapy [Hayes & Hofman, 2017], and incorporated a strong spiritual element [24].

Additional reference: Hayes, S.C.; & Hofmann, S.G. (2017). The third wave of cognitive behavioral therapy and the rise of process-based care. World Psychiatry 2017, 16(3), 245–246. https://doi.org/10.1002/wps.20442

Reviewer 3 Report

Reviewer notes IJERPH (ISSN 1660-4601)

Feb 2022

Promoting mental health and wellbeing in multicultural Australia: A collaborative regional approach.

Page 3

-The abbreviation CD is not defined before first appearing on line 101. There is a generational/technological limitation to knowledge of what this means.

-A similar issue is a brief explanation needed for your acronym USB.

Page. 8.

-Same issue for the acronyms IB and HS on this page.

Author Response

REVIEWER 3
Comments

Page 3
-The abbreviation CD is not defined before first appearing on line 101. There is a generational/technological limitation to knowledge of what this means.
-A similar issue is a brief explanation needed for your acronym USB.

Page. 8.
-Same issue for the acronyms IB and HS on this page.

Response

We have now spelt out all acronyms on their first use.